# Feed management practices used for dairy cows in confined dairies in Brazil

**Marcelo B. Abreu[1], Jessica M. V. Pereira[2], Virginia L. N. Brandao[3], Marcos Inacio Marcondes[4]\***

**1** Department of Animal Science, Universidade Federal de Viçosa, Viçosa, Minas Gerais, Brazil,
**2** Lawley's Inc, Tulare, California, United States of America, **3** Elanco US, Indianapolis, Idaho, United States of America, **4** William H. Miner Agricultural Research Institute, Chazy, New York, United States of America

\* mmarcondes@whminer.com

## Abstract

Despite the critical role that feeding and feed bunk management practices play in dairy herd performance, limited information is available on how these practices are implemented across different production levels in Brazilian confined dairy systems. We aimed to gather information on the producers 'reported feeding practices used on confined dairy operations and identify their relationship with average herd milk production. An electronic survey of 38 questions was emailed to 500 Brazilian dairy producers. Out of the 135 responses received, answers from 82 producers were included, and herds were ranked according to their 305-d milk production (kg) as low (LP; < 7,000; n = 27), medium (MP; 7,000–10,000; n = 35), and high production (HP; > 10,000; n = 20). The HP and MP herds had greater odds of evaluating particle size distribution, forage DM, TMR physically effective fiber NDF (peNDF), corn kernel processing, and having a water trough wash protocol and cooling system than LP herds. The HP herds had greater odds of grouping primiparous separated than LP herds. The feed refusals and frequencies of feed efficiency, feed bunk clean-up, and TMR peNDF were similar among herds. The HP producers reported a greater feeding frequency, feed push-up, and cleaning water troughs than MP and LP, respectively. Producers reported evaluating forage DM monthly and when a new silo was opened, regardless of milk production level. In conclusion, this survey demonstrates that producers of HP and MP herds showed a greater frequency of use of feeding management practices than those of LP herds. Furthermore, survey results can be used to develop and disseminate target information on feeding practices and feed bunk management in dairy operations.

## Introduction

Dairy production systems have undergone substantial transformation in recent decades, largely driven by the dissemination and adoption of technologies aimed at

**Data availability statement:** Data is available at Figshare: 10.6084/m9.figshare.31539448.

**Funding:** The author(s) received no specific funding for this work.

**Competing interests:** The authors have declared that no competing interests exist.

maximizing productivity [1]. In Brazil, this progress has been supported by key factors, including transition from pasture-based to confined systems, intensified genetic selection, improvements in cow comfort, and advances in nutritional management [2,3]. As a result, over the past 20 years, Brazilian milk production has increased by 59%, primarily driven by a 95% increase in productivity per cow, despite an approximately 19% reduction in the number of milked cows [1].

The widespread adoption of new technologies has been a key driver of progress in the dairy industry. Notably, even small- and medium-sized producers are increasingly adopting intensive feeding, management, and housing strategies that were previously characteristic of large-scale operations [4]. While these approaches can improve productivity and efficiency, they also introduce new challenges, particularly with respect to adoption, implementation, and maintaining consistency in feeding management practices.

In this context, especially in confined production systems the adoptions of feeding and management practices have been associated to affect cows feeding behavior and performance. Given the significant impact of management practices on the success of dairy operations, it is essential to provide the dairy industry with comprehensive information on the management strategies used on farms [5]. Consequently, it becomes crucial for dairy producers, especially those overseeing confined systems, to grasp the correlation between feed management and animal performance.

Although numerous studies have examined the effects of individual feeding strategies on milk yield and cow performance [6,7], there is a notable gap in the literature regarding the extent to which these strategies are adopted in commercial dairy operations. This lack of data limits producers' and advisors' ability to evaluate how specific feeding strategies relate to animal performance.

To the best of our knowledge, no study has systematically summarized producer-reported feeding management practices and evaluated their association with herd-level milk production. Therefore, the objective of this study was to describe the most commonly reported feeding practices used in high milk production pens on dairy farms and to determine whether specific feeding management strategies are associated with herd-level milk production (**HLMP**).

## Materials and methods

The ethics committee of the Universidade Federal de Viçosa reviewed the study protocol and waived the requirement for ethical approval, as there were no animal or human subjects and all surveys were anonymous. An electronic cross-sectional survey was developed to gather information from feed practices and nutritional management on dairy operations located in southern and southeastern Brazil. The survey targeted dairy producers located in regions that collectively account for 66.6% of the country's total milk production in the country [1]. The survey was written in Portuguese, and an introductory letter was included to explain its purpose. A total of 500 electronic survey invitations were once distributed to Brazilian dairy producers via email and social media platforms (e.g., Instagram). Producer contact information

was obtained through the dairy industry and consulting nutritionists. On social media, invitations were sent via direct messaging. The survey targeted only dairy producers who housed cows in confinement systems, including free-stall barns, compost barns, dry lots, and tie stalls (Fig 1).

Informed consent was obtained electronically prior to participation and was mandatory for accessing the survey. Only individuals who provided consent were allowed to proceed, and responses were recorded exclusively from consenting participants; individuals who did not provide consent were unable to continue, and no data were collected. Participation was voluntary, responses were analyzed anonymously and in aggregate form, and no personal identifying information was required. The study did not involve minors.

The survey consisted of 38 questions, including both open-ended and multiple-choice formats, with no mandatory responses and a single-response option per question. The questionnaire was organized into four sections: (1) herd demographics, (2) total mixed ration (TMR) preparation and feedstuff evaluation, (3) feed bunk management practices, and (4) high-producing cow management. In the introduction, respondents were informed that the questions explicitly referred to cows housed in the pen with the highest milk production on their farm. The original survey was written in Portuguese and later translated into English solely for the purpose of manuscript submission to this journal (S1 Table). The questionnaire was developed using electronic survey software (Qualtrics, Provo, UT), and personalized survey links were generated to minimize the risk of duplicate responses. A pilot version of the survey was administered to three dairy producers and four nutritionists to assess the clarity of questions and the overall organization of the content. Based on their feedback, minor revisions were made. On average, respondents took approximately 10–15 minutes to complete the questionnaire. Survey responses were collected between December 2020 and February 2021.The survey achieved a response rate of 27.6%, with 138 responses received from 500 invitations. Of these, 21 blank responses and 35 incomplete surveys—defined as questionnaires that were started but not completed, then excluded from the dataset. The final analysis was conducted using the remaining 82 complete responses. The overall survey margin of error was 7.1%, based on a 95% confidence level [8].

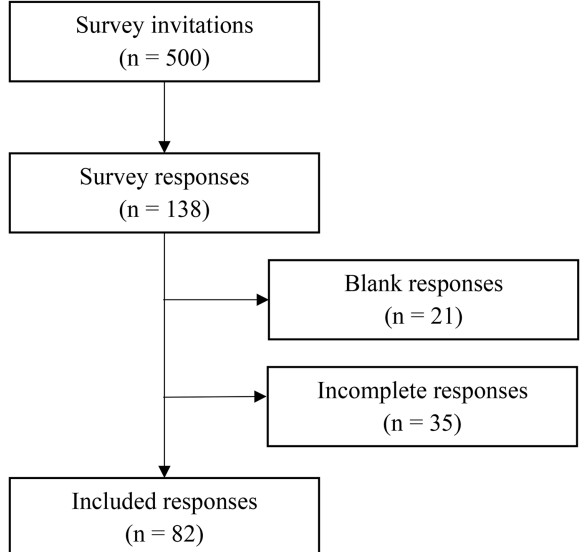

**Fig 1. Flow diagram illustrating the survey process, from initial invitation and eligibility screening to the final analytical dataset, which included 82 completed responses.**

## Experimental design and statistical analysis

Survey responses were exported from the electronic survey software into Microsoft Excel® (Version 2302) for data management and cleaning. Responses were reviewed for coherence and completeness. Twelve surveys that were submitted entirely blank were excluded, and any conflicting or inconsistent responses were omitted from the analysis. The normality of non-dichotomous variables was assessed using the UNIVARIATE procedure in SAS (SAS Institute Inc., Cary, NC). Chi-squared tests were performed to evaluate whether the observed frequency distributions matched expected values. Descriptive statistics were calculated using the PROC MEANS and PROC FREQ procedures in SAS 9.4 (SAS Institute Inc., Cary, NC). Reported percentages refer only to respondents who reported practicing the respective management strategies. Herds were classified into three milk production categories—low (LP; < 7,000 L/cow per 305 days), medium (MP; 7,000–10,000 L/cow), and high (HP; > 10,000 L/cow)—based on 305-day average milk yield per cow as reported by herd nutritionists. This classification reflects standard benchmarks used in the region to describe small, medium, and large-scale dairy operations. Herd-level average milk production was used for classification, as cows are typically moved between pens as production declines. Because milk yield did not follow a normal distribution across herds, group sizes varied among the production categories. Feeding practice questions with binary ("yes" or "no") responses were dichotomized for analysis. Odds ratio (OR) comparisons were conducted to evaluate associations between herd milk production level (LP, MP, HP) and feeding practices. Three pairwise comparisons were made using Poisson regression via the GENMOD procedure in SAS 9.4. Low-production herds (LP) were used as the reference category when estimating ORs for MP and HP herds, and MP herds served as the reference in comparisons with HP herds.

## Results

Descriptive statistics for each herd production group are presented in Table 1. The LP, MP, and HP herd groups consisted of 27 (33%), 35 (42%), and 20 (24%) herds, respectively. It is important to note that the number of respondents represents only a small fraction of the Brazilian dairy producer population, which the national number of producers totaled approximately 1.1 million in 2019 [9]. Among all categorical variables analyzed, only the frequency of forage dry matter (DM) evaluation and the presence of a water trough cleaning protocol deviated significantly from the expected distribution in the chi-squared test. These two variables are discussed in detail in the following sections.

The average number of lactating cows per herd was 171 ± 294. In terms of breed distribution, 63% of the herds were Holstein (n = 52), 27% were Holstein × Gyr crosses (n = 22), and 10% were Jersey (n = 8). Regarding housing systems, 51% of the herds housed their cows in compost barns (n = 42), 29% in dry lots (n = 24), and 20% in free-stall systems (n = 16). The surveyed herds were housed in free-stall barns, compost-bedded pack barns, or dry lots. Although the survey did not include a detailed assessment of housing system characteristics, a comprehensive description of housing systems in the state of Minas Gerais has been previously reported by [2]. In brief, cows in free-stall systems were group-housed and provided with individual bedding composed of either sand or mattresses, with access to a central alley. In compost-bedded pack barns, cows had continuous access to a shared bedding area averaging 15 m² per animal, typically made of wood shavings or coffee hulls, with fans installed above the bedding to enhance ventilation. In open dry-lot systems, cows were managed in large uncovered areas averaging 55 m² per animal, with feeding areas consisting of non-covered concrete feed bunks and a minimum of 4.0 m² of shaded space per cow. Most herds were composed of a single breed; however, two herds included more than one breed. In both of these cases, approximately 85% of the cows were Holstein × Gyr crossbreds and were thus classified as Girolando herds for this study.

The reported 305-day milk yields varied across herds: 17% reported production below 6,000 kg (n = 14); 16% between 6,000 and 7,000 kg (n = 13); 13% between 7,000 and 8,000 kg (n = 11); 12% between 8,000 and 9,000 kg (n = 10); 17% between 9,000 and 10,000 kg (n = 14); 9% between 10,000 and 11,000 kg (n = 7); 6% between 11,000 and 12,000 kg (n = 5); 9% between 12,000 and 13,000 kg (n = 7); and 1% above 13,000 kg (n = 1).

**Table 1. Descriptive characteristics of dairy farms according to the three groups of herd milk production level and the number of respondents within category in parenthesis.**

| | Herd milk production level | | | Overall |
| --- | --- | --- | --- | --- |
| | LP[1] (n = 27) | MP[2] (n = 35) | HP[3] (n = 20) | |
| **Lactating cows, n** | | | | |
| Mean | 135 | 150 | 273 | 175 |
| Median | 50 | 110 | 65 | – |
| Maximum | 1950 | 600 | 1200 | – |
| Minimum | 10 | 23 | 10 | – |
| **Herd 305d milk production** | | | | |
| Mean | 5,981 | 8,586 | 11,625 | – |
| Median | 5,500 | 8,500 | 11,500 | – |
| Maximum | 6,500 | 9,500 | 13,000 | – |
| Minimum | 6,000 | 7,000 | 10,000 | – |
| **Herd breed, n herds, % (n)** | | | | |
| Holstein | 12 (33) | 21 (34) | 19 (95) | 52 (63) |
| Holstein x Gyr | 9 (45) | 12 (60) | 1 (5) | 22 (27) |
| Jersey | 6 (22) | 2 (6) | – | 8 (10) |
| **Housing system, n herds, % (n)[4]** | | | | |
| Compost Barn | 8 (30) | 22 (63) | 12 (60) | 42 (51) |
| Dry lot | 16 (58) | 7 (20) | 1 (5) | 24 (29) |
| Free stall | 3 (12) | 6 (17) | 7 (35) | 16 (20) |
| **Milkings, n herds, % (n)[4]** | | | | |
| 2 | 21 (79) | 16 (46) | 8 (40) | 45 (55) |
| 3 | 6 (21) | 19 (54) | 12 (60) | 37 (45) |

[1] Low production herd (< 7,000 L/cow).

[2] Medium production herd (7,000–10,000 L/cow).

[3] High production herd (> 10,000 L/cow).

[4] Housing system and number of milkings in the high milk production pens.

Survey responses were primarily from the state of Minas Gerais (n = 55; 67%), followed by Santa Catarina (n = 9; 11%), Rio Grande do Sul (n = 7; 9%), Paraná (n = 5; 6%), São Paulo (n = 4; 5%), and Espírito Santo and Rio de Janeiro (n = 1 each; 1%). Among the 82 respondents, 74 (90%) reported working with a nutritionist (Table 1).

## Total mixed ration preparation

All surveyed farms utilized a total mixed ration (TMR) feeding system. Mixer wagons are used to thoroughly blend feed ingredients, ensuring a well-homogenized diet is delivered along the feed bunk [6]. A higher proportion of HP and MP herds reported using mixer wagons compared to LP herds, with only 58% of the latter indicating their use. However, the odds ratio for using a mixer wagon was significantly greater only for MP herds compared to LP herds (Fig 2). Among the herds using mixer wagons, vertical mixers were reported in 75% of HP herds, 58% of MP herds, and 60% of LP herds. In addition, corn silage and Tifton hay were the most commonly used forages among participating farms. Ground corn, soybean meal, whole cottonseed, and citrus pulp were the most frequently included ingredients in the concentrate.

Overall, respondents reported adding ingredients to the mixer wagon in the following order: forages (hay and silages), concentrates, by-products, and finally minerals and vitamins—regardless of HLMP. Similarly, reported TMR mixing time ranged from 5 to 10 minutes after the inclusion of the final ingredient, with no variation observed across HLMP groups (Table 2).

 

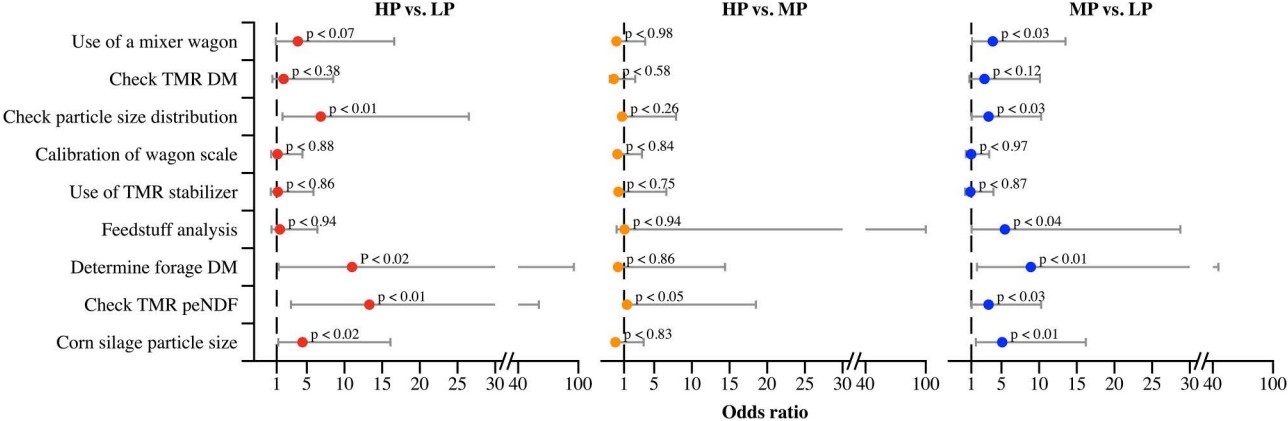

**Fig 2. Odds ratios for total mixed ration preparation and feed evaluation practices reported by Brazilian dairy producers.** Herds were categorized based on 305-day milk yield as follows: high production (HP; >10,000 L/cow), medium production (MP; 7,000–10,000 L/cow), and low production (LP; <7,000 L/cow). Comparisons were made using LP herds as the reference group for HP (pink dots) and MP (blue dots) herds, and MP herds as the reference group for comparisons with HP herds (yellow dots). Each dot represents the odds ratio, and gray bars indicate the 95% confidence interval. The dashed line denotes the reference group (odds ratio = 1). peNDF = physically effective neutral detergent fiber.

The odds ratio for evaluating TMR DM was not significantly associated with HLMP (Fig 2). Most producers, regardless of production level, reported targeting a TMR moisture content between 50% and 55%, and approximately 75% indicated this as their ideal DM range.

Wagon mixer scale calibration practices were also not associated with HLMP (Fig 2). Only 42% of producers reported calibrating the wagon scale at least once per year (Table 2). Additionally, the use of TMR stabilizers was not influenced by HLMP (Fig 2), with just 15% of respondents indicating their use. Lastly, the odds ratios for TMR preparation practices were similar between HP and MP herds.

### Feeding practices in high-production pens

In the present survey, HP herds numerically reported feeding cows three times per day more often, while MP and LP herds most commonly reported feeding twice per day (Fig 3A). The OR for feeding cows after milking was not associated with HLMP (Fig 4). Similarly, HLMP was not associated with feed push-up frequency (Fig 4). However, 41% of HP herds reported conducting five or more feed push-ups per day, 47% of MP herds reported three to four push-ups per day, and 50% of LP herds reported one to two push-ups per day (Fig 3B).

In the present study, HP, MP, and LP herds reported similar targets for feed delivery. Most producers, regardless of HLMP, indicated aiming for approximately 5% feed refusals (Fig 3C). The OR for evaluating FE was higher in HP and MP herds compared to LP herds, although no difference was observed between HP and MP herds (Fig 4). Across all HLMP groups, 83% of producers reported evaluating FE on a monthly basis (Fig 3D).

HP and MP herds showed higher ORs for having water trough cleaning protocols compared to LP herds, but all groups had similar ORs for feed bunk cleaning (Fig 4). Additionally, HP producers reported cleaning water troughs seven times per week more frequently than MP and LP producers (Fig 3F). The provision of employee training in feed management was not associated with HLMP (Fig 4).

### High-production pens management

The most commonly reported feed bunk space in HP pens ranged from 70 to 80 cm, with no notable differences across HLMP groups (Fig 5A). A stocking density of 100% for high-producing lactating cows was reported by 56% of HP herds,

**Table 2. Descriptive characteristics of feed practices of dairy farms according to the three groups of herd milk production level and the number of respondents within category in parentheses.**

| Item | Herd milk production level, % | | |
|---|---|---|---|
| | LP (n = 27)[1] | MP (n = 35)[2] | HP (n = 20)[3] |
| **Have nutritionist, n herds, % (n)[4]** | 20 (75) | 35 (100) | 19 (95) |
| **Nutritionist visit frequency,% (n)** | – | – | – |
| Weekly | 36 (5) | 19 (6) | 24 (4) |
| Biweekly | – | 47 (15) | 12 (2) |
| Monthly | 50 (7) | 34 (11) | 59 (10) |
| >Monthly | 14 (2) | – | 6 (1) |
| **Mixing type[4],** | | | |
| Horizontal | 60 (3) | 42 (5) | 25 (1) |
| Vertical | 40 (2) | 58 (7) | 75 (3) |
| **Mixing time[5], min (n)** | | | |
| ≤5 | 17 (1) | 36 (4) | 50 (3) |
| <5 to ≤ 10 | 50 (3) | 36 (4) | 17 (1) |
| <10 to ≤ 15 | 33 (2) | 18 (2) | 17 (1) |
| <15 to ≤ 20 | – | 9 (1) | 17 (1) |
| **TMR moisture goal[6], % (n)** | | | |
| <50 | 5 (1) | 12 (3) | 7 (1) |
| 50 ≤ to ≤55 | 95 (19) | 88 (22) | 93 (13) |
| >55 | – | – | – |
| **Particle size distribution, frequency (n)** | | | |
| Weekly | – | – | 9 (1) |
| Biweekly | 14 (1) | 27 (4) | 18 (2) |
| Monthly | 57 (4) | 40 (6) | 45 (5) |
| >Monthly | 29 (2) | 33 (5) | 27 (3) |
| **Scale calibration, frequency/yr (n)** | | | |
| 1x | – | 25 (3) | 29 (2) |
| 2x | 43 (3) | 25 (2) | 29 (2) |
| ≥3 | 57 (4) | 50 (6) | 43 (3) |

[1] Low production herds (< 7,000 L/cow).

[2] Medium production herds (7,000–10,000 L/cow).

[3] High production herds (> 10,000 L/cow).

[4] Considering only those using wagon mixer.

[5] Time after last ingredient load.

[6] Total mixed ration.

50% of MP herds, and 45% of LP herds (Fig 5B). Overstocking (>100% stocking density) was reported by 19% of HP, 4% of MP, and 5% of LP producers. When asked about the frequency of moving cows into higher-production pens, 52.9% of HP producers reported doing so daily, whereas 31.0% of MP and 39.3% of LP producers reported moving cows on a bi-weekly basis.

In the current study, HP herds were more likely than LP herds to group primiparous cows separately from multiparous cows (Fig 4). Additionally, HP and MP producers more frequently reported maintaining this separation throughout the entire lactation, whereas LP producers more commonly separated primiparous cows only during the close-up

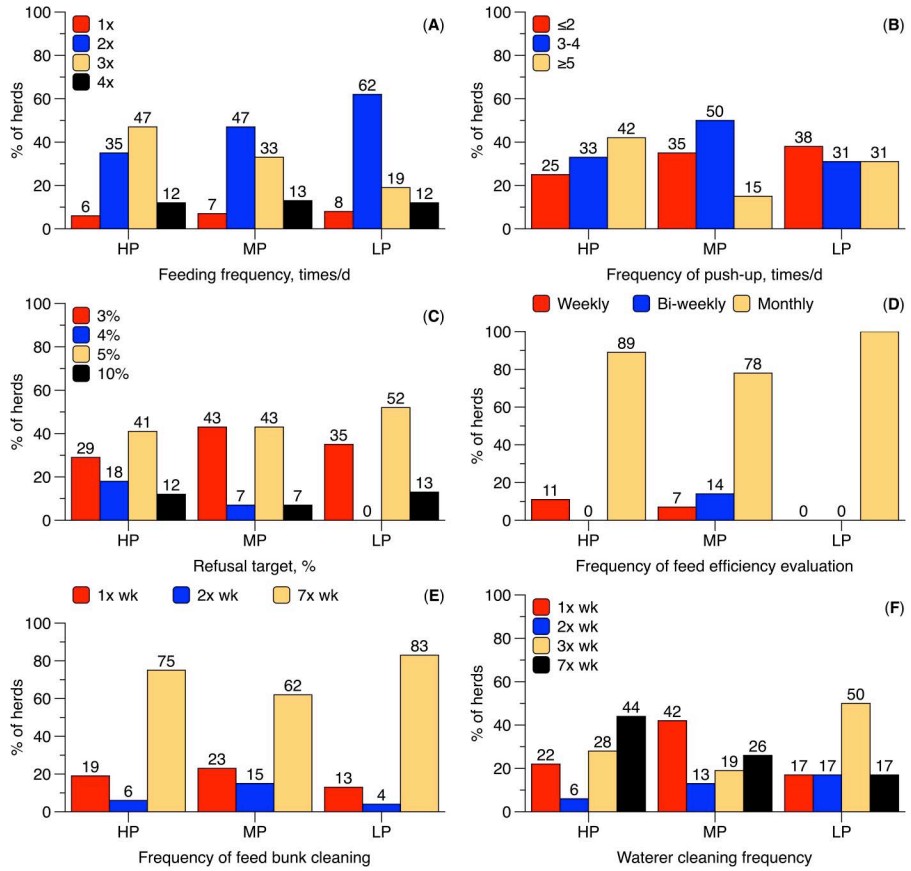

**Fig 3. Frequency of feeding management practices reported by Brazilian dairy producers.** Herds were categorized based on 305-day milk yield as follows: high milk production (HP; > 10,000 L/cow), medium milk production (MP; 7,000–10,000 L/cow), and low milk production (LP; < 7,000 L/cow). Panels show the reported frequency of the following practices: cow feeding **(A)**, feed push-up **(B)**, feed refusal targets **(C)**, monitoring of feed efficiency **(D)**, feed bunk cleaning **(E)**, and water trough cleaning **(F)**. Percentages displayed above the bars represent the proportion of respondents selecting each option.

period (Fig 5C). HP and MP herds also showed a higher OR for having a cooling system compared to LP herds, with no differences observed between HP and MP groups (Fig 4).

### Diet evaluation

HP and MP herds showed greater ORs for determining forage DM and evaluating corn processing compared to LP herds (Fig 2). MP producers also reported submitting feed samples for laboratory analysis more frequently than HP producers, with LP herds reporting the lowest frequency (Fig 5A). Overall, the majority of producers—regardless of HLMP—indicated receiving monthly visits from a nutritionist (Table 2).

HP and MP herds showed greater ORs for determining forage DM and evaluating corn processing compared to LP herds (Fig 2). MP producers also reported submitting feed samples for laboratory analysis more frequently than HP producers, with LP herds reporting the lowest frequency (Fig 6A). Monthly forage DM assessment was most common among MP herds (58%), followed by HP herds (38%) (Fig 6B). MP producers also reported a higher frequency of corn processing evaluation than both HP and LP herds (Fig 6D). HP and MP herds also had greater ORs for evaluating TMR particle size

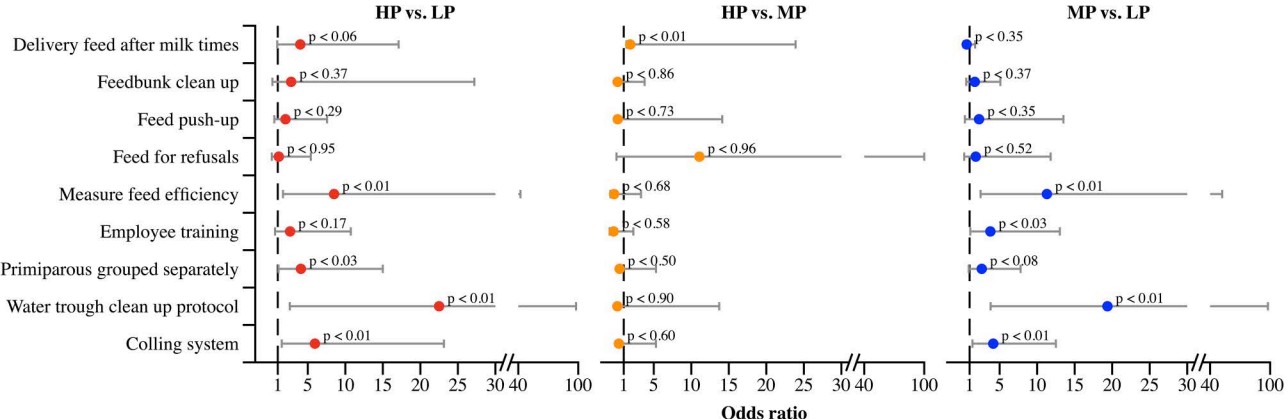

**Fig 4. Odds ratios for feeding and management practices reported by Brazilian dairy producers.** Herds were categorized based on 305-day milk yield as follows: high production (HP; > 10,000 L/cow), medium production (MP; 7,000–10,000 L/cow), and low production (LP; < 7,000 L/cow). LP herds served as the reference group for comparisons with HP (pink dots) and MP (blue dots) herds, while MP herds were the reference group for comparisons with HP herds (yellow dots). Dots represent odds ratios, gray bars indicate 95% confidence intervals, and the dashed line denotes the reference level (odds ratio = 1).

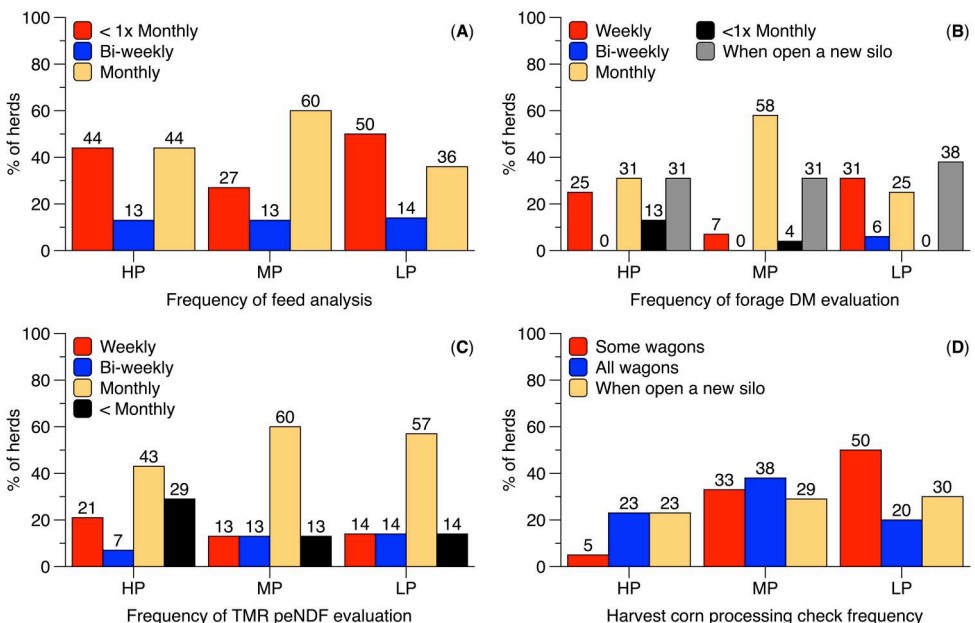

**Fig 5. Frequency of selected management practices reported by Brazilian dairy producers, categorized by 305-day milk yield: high production (HP; > 10,000 L/cow), medium production (MP; 7,000–10,000 L/cow), and low production (LP; < 7,000 L/cow).** Panels show reported feed bunk space **(A)**, stocking density for high-producing cows **(B)**, and use of dedicated pens for primiparous cows **(C)**. Percentages displayed above the bars represent the proportion of respondents selecting each option.

distribution and peNDF compared to LP herds, with similar ORs observed between HP and MP herds (Fig 2). The most commonly reported frequency of TMR peNDF evaluation was once per month, regardless of HLMP (Fig 6C). No other feed practices differed significantly between HP and MP herds (Figs 1 and 2).

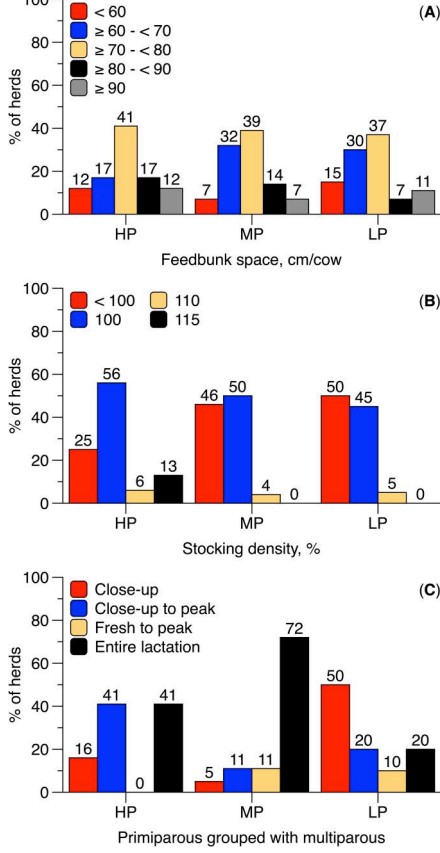

**Fig 6. Frequency of feeding practices reported by Brazilian dairy producers.** Herds were categorized based on 305-day milk yield: high milk production (HP; >10,000 L/cow), medium milk production (MP; 7,000–10,000 L/cow), and low milk production (LP; <7,000 L/cow). Panels represent the frequency of cow feeding **(A)**, feed push-up **(B)**, feed refusal targets **(C)**, monitoring of feed efficiency **(D)**, feed bunk cleaning **(E)**, and water trough cleaning **(F)**. Percentages above the bars indicate the proportion of respondents selecting each option.

## Discussion

### Total mixed ration preparation

The lower adoption of mixer wagons by LP producers compared to MP and HP herds may reflect the smaller herd sizes and limited economic justification for such an investment [8]. Despite this, the majority of producers across all HLMP categories reported targeting a TMR DM between 50 and 55%, which falls within the recommended range of 45–60% to maximize feed intake [6,10]. This suggests that, overall, herds included in this study may be at lower risk for subacute ruminal acidosis due to reduced sorting against long particles [11,12] or the adverse effects on DMI associated with excessively dry or wet TMRs [13].

Accurate calibration and maintenance of mixer wagon scales are essential to ensure proper diet formulation and uniform feed delivery. Previous research has shown that discrepancies between formulated and delivered rations often stem from ingredient loading errors and lack of scale calibration [14,15]. In this study, the frequency of scale calibration was similar across HLMP groups, with most producers reporting monthly calibration. Given the significant financial investment feed represents on dairy farms, proper scale calibration is not only a matter of operational precision but also a key factor in animal performance and accurate estimation of feed efficiency [14].

Interestingly, only 15% of producers reported using TMR stabilizers, despite their proven effectiveness in preventing heating and spoilage under warm conditions. High temperatures favor mold growth, which can compromise feed quality [16] and reduce intake [10]. Prior research in Brazil has demonstrated that organic acid-based stabilizers can reduce TMR temperature and improve both DMI and fat-corrected milk yield [17]. These results suggest an opportunity for wider adoption of TMR stabilizers in Brazil, particularly when economically viable. Moreover, producers and nutritionists should consider additional strategies to improve aerobic stability, such as proper silo design, efficient harvesting and packing methods, and the use of heterofermentative inoculants.

## Feeding practices in high-production pens

Feed delivery is widely recognized as one of the most effective nutritional management strategies to stimulate dry matter intake and reduce feed sorting [18,19], which may partially explain the higher milk yields observed in HP herds compared to MP and LP herds. Delivering feed post-milking has also been linked to a lower incidence of mastitis, as it encourages cows to remain standing, giving the teat sphincter sufficient time to close [20]. In a cross-sectional study, herds that provided fresh feed immediately after morning and evening milkings showed a lower incidence of clinical mastitis compared to herds that did not follow this practice [21]. Similarly, cows that remained standing for 40–60 minutes after milking were found to have a lower risk of intramammary infection than those lying down within the first 40 minutes [22]. Although the OR for feeding cows post-milking was not associated with HLMP in this study (Fig 2), it is important to note that data on clinical mastitis and somatic cell counts were not collected.

Feed push-up is another key management strategy used to ensure consistent feed availability. A greater frequency of push-ups has been associated with reduced feed sorting, increased eating time, and a more even distribution of feeding behavior throughout the day [23]. However, results from previous studies are mixed. For instance, one study found that increasing feed push-up frequency from 3 to 5 times per day did not impact feed sorting, lying time, or milk yield and composition [24]. On the other hand, another study reported a positive association between milk yield and push-up frequency [25], although the cows in that study were housed in tie-stall barns, which may limit the applicability of the findings to freestall or compost-bedded systems due to differences in social dynamics and feeding behavior.

It is common for lactating cows to be offered slightly more feed than their predicted DMI to allow for flexibility in intake. The duration of feed availability has been linked to feed sorting behavior and milk production outcomes [26]. Although herds in this study had similar feed delivery targets, consistent management of TMR refusals should be encouraged to reduce feed waste and improve feeding precision. Feed efficiency remains a widely used metric to evaluate production efficiency. Although HP and MP had greater OR for evaluating feed efficiency among HLMP groups, monthly evaluation was the most frequently reported practice across all herd categories. This could suggest that herds with a high frequency FE evaluation may better detect deviations from target efficiency and improve productivity through timely dietary and management adjustments.

## High-production pens management

Extensive literature has demonstrated differences in nutritional requirements and feeding behavior between primiparous and multiparous cows, supporting the practice of housing them separately [27,28]. For example, [29] reported a 1.59 kg/day increase in milk yield when primiparous cows were housed in separate pens. The greater number of lactating cows in HP herds compared to LP herds may facilitate and justify the separation of these groups. This observation aligns with findings from [30], which indicated that grouping decisions are often influenced by overall herd size. While pen availability, facility design, and labor considerations can limit the implementation of this strategy, producers should be encouraged to evaluate feeding behavior and nutritional needs based on lactation stage as a foundation for effective grouping strategies.

Environmental factors, particularly high temperatures and humidity common in tropical climates, are linked to reduced performance and economic losses in dairy cows. A review by [31] showed that the use of heat abatement strategies

significantly improves both milk production and reproductive outcomes. Although this survey did not assess the reasons for the lack of cooling systems, the lower adoption rate in LP herds may be attributed to their greater reliance on dry-lot housing systems (Table 1), which are more commonly associated with less intensive production. Additionally, many LP herds in this study used Holstein x Gyr crossbreds, a breed known for its increased resilience to heat stress. This may further reduce the perceived need for investing in active cooling infrastructure.

High-producing dairy cows have highd nutrient demands, requires precise diet formulation and delivery. Nutrient deficiencies can compromise milk yield, body condition, reproductive performance, and immune function [32]. Regular evaluation of feed ingredients, including the composition of the TMR and particle size distribution, is crucial for precision feeding. While the relationship between HLMP and feedstuff evaluation varied across practices, farms with more comprehensive ingredient assessments likely have greater capacity to meet nutritional targets and improve performance outcomes.

Providing cows with adequate particle size and physically effective fiber is critical to maintaining rumen health and optimizing milk production [33]. Studies have shown that forage particle size can change substantially during TMR preparation and delivery [23,34]. In this study, HP and MP herds reported more frequent assessments of particle size and peNDF, while LP herds lagged in this practice. This highlights the importance of routine monitoring by producers and nutritionists to ensure consistent TMR quality and to support improved animal performance through well-managed feeding systems.

## Conclusion

In the presente study, HP and MP herds more frequently and intensively implemented feeding management practices, indicating a higher level of technical knowledge application. Improving feeding strategies in Brazilian dairies requires effective knowledge transfer from academic and technical professionals to producers. Skilled consultants are crucial in helping to implement practices that enhance nutritional precision and performance. Although the study provides valuable insights, the sample size limits the ability to make broad generalizations. Future research should include a greater number of respondents from diverse regions to develop more informed, location-specific recommendations.

## Supporting information

**S1 Table. Survey questions with their respective answers' options.**
(DOCX)

## Acknowledgments

The authors are grateful to all producers who contributed to the study. The authors thank the Conselho Nacional de Desenvolvimento Cientifico for financial support during the study.

## Author contributions

**Conceptualization:** Marcos Inacio Marcondes.

**Formal analysis:** Marcelo B. Abreu, Marcos Inacio Marcondes.

**Investigation:** Marcelo B. Abreu, Jessica M. V. Pereira.

**Methodology:** Marcelo B. Abreu, Jessica M. V. Pereira, Marcos Inacio Marcondes.

**Project administration:** Marcos Inacio Marcondes.

**Supervision:** Marcos Inacio Marcondes.

**Visualization:** Virgínia L. N. Brandão.

**Writing – original draft:** Marcelo B. Abreu, Jessica M. V. Pereira, Virgínia L. N. Brandão, Marcos Inacio Marcondes.

**Writing – review & editing:** Marcelo B. Abreu, Marcos Inacio Marcondes.

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
