## [Decision Letter · Decision Letter 0]

12 Oct 2025

Dear Dr.  Marcondes,

We look forward to receiving your revised manuscript.

Kind regards,

Marcia Saladini Vieira Salles

Academic Editor

PLOS ONE

Journal Requirements:

2. You indicated that ethical approval was not necessary for your study. We understand that the framework for ethical oversight requirements for studies of this type may differ depending on the setting and we would appreciate some further clarification regarding your research. Could you please provide further details on why your study is exempt from the need for approval and confirmation from your institutional review board or research ethics committee (e.g., in the form of a letter or email correspondence) that ethics review was not necessary for this study? Please include a copy of the correspondence as an ""Other"" file.

Additional Editor Comments:

A well-designed cross-sectional study describing feed management practices on dairy farms in Brazil. The findings are useful for extension and benchmarking purposes and support the need for greater adoption of precision practices by higher-producing herds.

Introduction

L43 to 55: Please check if this statement is valid, it seems to me to be a very high percentage of increase in milk production per cow for a one-year outlook. ‘As a result, dairy production has grown substantially worldwide, where milk production per cow increased by 88% from 2017 to 2018 [4].’

L55 to 56: Please set this information to a more recent value, it has been almost five years. ‘In addition, from 2006 to 2017, fluid milk production increased by 45% country-wide and reached 35.4 billion L in 2020 [5].

The citations for key information contained in the introduction to the surveys on the property situation in Brazil are over 10 years old. It would be important to update this information.

M&M

Suggestion to include a flowchart in M&M (n invitations → n responses → exclusions → n final per analysis).

Results

L152 to 156: This part belongs to the materials and methods section.

L185: The entire description in this paragraph belongs to table 2.

Figures

Figure 1: It would be more didactic and easier for readers to understand if the authors presented the sequence of comparisons first HPxMP, followed by HPxLP, and MPxLP.

Tables

Table 1: Herd ‘Her milk production level’

Conclusion

The conclusion should be a little more straightforward. Remove the initial section summarizing the results; this has already been presented in the respective section.

References

L486: Custom

Reviewer's Responses to Questions

**Comments to the Author**

1. Is the manuscript technically sound, and do the data support the conclusions?

Reviewer #1: Partly

Reviewer #2: Yes

Reviewer #3: Yes

2. Has the statistical analysis been performed appropriately and rigorously?

Reviewer #1: I Don't Know

Reviewer #2: Yes

Reviewer #3: Yes

3. Have the authors made all data underlying the findings in their manuscript fully available?

Reviewer #1: No

Reviewer #2: Yes

Reviewer #3: Yes

4. Is the manuscript presented in an intelligible fashion and written in standard English?

Reviewer #1: Yes

Reviewer #2: Yes

Reviewer #3: No

Reviewer #1: Introduction- The Introduction is a jumble of sentences without a clear logic. What is the study about? What is the context that justifies the research? What is the research intended to answer? The Introduction should be rewritten to clarify the context, justification, state of the art, and objective of the study.

Line 53-54- "As a result, dairy production has grown substantially worldwide, where milk production per cow increased by 88% from 2017 to 2018." Where did milk production per cow increase? It's not possible that this happened worldwide.

Line 55-56- "In addition, from 2006 to 2017, fluid milk production increased by 45% country-wide and reached 35.4 billion L in 2020." Which country are you referring to?

Material and Methods- The description of the research methodology is incomplete. For example, why was the survey only conducted in the southern and southeastern regions of Brazil? These regions are vast. Where were the producers who received the questionnaires concentrated (state and region)? Were the state and region factors considered in the data analysis? What was the process for developing the 38 questions in the questionnaire (based on the literature)? Was the questionnaire sent only once, or were there multiple attempts?

The content between lines 108 and 124 is the result of the questionnaires. Why are they included in the materials and methods section?

Line 39-140- This classification reflects standard benchmarks used in the region to describe small, medium, and large-scale dairy operations." What is the reference for this classification?

Results- The results are discussed superficially, descriptively, without critical analysis of the findings and their meaning.

Lines 153-155- 138 responses , 21 blank responses and 33 incomplete surveys were excluded from the dataset. The final

analysis was conducted using the remaining 82 complete responses. The correcte number is 84

Lines 157-159. It is important to note that the number of respondents represents only a small fraction of the Brazilian dairy producer population, which totaled approximately 1.1 million in 2019." But this is the total number of producers in Brazil. It wasn't the study's target audience, but rather only those in confined systems.

Reviewer #2: The manuscript ‘Feed Management Practices Used for High-Producing Dairy Cows’ aligns well with the scope of PLOS ONE and provides valuable insights into management practices in Brazilian intensive dairy systems. However, minor revisions are required before the manuscript can be accepted. The following suggestions are offered to improve clarity and rigor:

Title: The title could be made more informative by including the country, at least, to provide greater precision. Dairy intensive systems in Brazil differ from those in other regions and have unique characteristics, as discussed in the manuscript.

Abstract:

L31: What was the criterion for selecting those 500? Please add a brief explanation to clarify.

Introduction:

L54: Review this statement.

L56: Please provide an updated value for milk production in Brazil.

L58-59: Please revise this sentence, as the phrasing currently resembles AI-generated text format. Ensure it aligns with a natural, academic writing style (review throughout the manuscript).

Lines 73-76: The stated objective seems too narrow. Based on the manuscript, the study does not focus solely on high milk production pens but includes herds across low, medium, and high production levels. Consider revising the objective to reflect the full scope of the study.

Material and methods:

Start the section with approval from the Ethics Committee.

L109: This information is duplicate. Please review.

L110-124: Please review this text. In my opinion, it does not belong in the Materials and Methods section. Consider moving it to the Discussion or removing it.

Results:

L164: Review last column format.

L166: Standardize with others.

L169: I did not understand.

L186-187: Blank lines.

L191: A reference is not needed if this information comes from the present study.

L224-229: Review the use of “;” and “.”.

L236: No not use “numerically”.

Discussion:

It would be interesting to discuss the survey’s representativeness in terms of herd size and production level relative to the Brazilian dairy producer population.

L344: Blank line.

L355-358: Avoid conclusion statements in each section.

L376: Please rephrase.

L387: Review to improve readability.

L408-410: Please review the text.

This result also deserves more discussion and implication “Across all HLMP groups, 83% of producers reported evaluating FE on a monthly basis”.

L426-440: I suggest removing this section from the text, as the manuscript is already concise and includes a conclusion section.

Conclusion:

L444: No dot use methods or results statements in this section. Be objective.

Acknowledgements:

The Financial Disclosure brings “The author(s) received no specific funding for this work.” . However, they acknowledge “Conselho Nacional de Desenvolvimento Cientifico” for financial support during the study. Please clarify.

Reviewer #3: The title should be "Feed management practices used for high-producing dairy cows in Brazil"

The manuscript provides important and up-to-date information. However, starting with the title, it should be clear that it refers to dairy systems in Brazil.

I recommend having all the manuscript writing and grammar reviewed by a native English speaker. As an example on the first page, there are more appropriate terms, for "furnish" such as "provide" on line 62, or "strategies employed on farms" such as "practices applied on farms" on line 63, and so on throughout the text.

There is also incorrect information such as "As a result, dairy production has grown substantially worldwide, where milk production per cow increased by 88% from 2017 to 2018" line 53;

It should be clear in phrases like " In addition, from 2006 to 2017, fluid milk production increased by 45% country-wide and reached 35.4 billion L in 2020", especially in the introduction, that you are referring to Brazil.

**Do you want your identity to be public for this peer review?** For information about this choice, including consent withdrawal, please see our Privacy Policy

Reviewer #1: No

Reviewer #2: No

Reviewer #3: **Yes:** Collao-Saenz Edgar A

---

## [Author Response · Author response to Decision Letter 1]

6 Feb 2026

Journal Requirements

1. Style and file name requirements were carefully checked and verified.

2. All required statements were included in the attached Ethics section.

3. Informed consent was obtained electronically prior to participation and was required to access the survey. Only individuals who provided consent were able to proceed, and data were collected exclusively from consenting participants. Participation was voluntary, responses were analyzed anonymously and in aggregate, no personally identifiable information was collected, and the study did not involve minors

4. All data were collected anonymously, with no access to personally identifiable information at any stage of the research process. Because the data were fully anonymized at the time of collection and individual identification was not possible, the ethics committee did not require an informed consent waiver. A letter from the Ethic committee from the Universidade Federal de Viçosa was attached to this submission for clarification.

5. Data will be made available upon acceptance of the manuscript.

6. Included as requested.

Introduction

L43 to 55: Please check if this statement is valid, it seems to me to be a very high percentage of increase in milk production per cow for a one-year outlook. ‘As a result, dairy production has grown substantially worldwide, where milk production per cow increased by 88% from 2017 to 2018 [4].’

R. The information was revised, and the Introduction was rewritten to provide a clearer and more accurate description of the study purpose.

L55 to 56: Please set this information to a more recent value, it has been almost five years. ‘In addition, from 2006 to 2017, fluid milk production increased by 45% country-wide and reached 35.4 billion L in 2020 [5].

R. The information has been revised, and the Introduction was rewritten to provide a clearer and more accurate description of the study purpose.

The citations for key information contained in the introduction to the surveys on the property situation in Brazil are over 10 years old. It would be important to update this information.

R. The citations were updated to include more recent references.

M&M

Suggestion to include a flowchart in M&M (n invitations → n responses → exclusions → n final per analysis).

R. A survey flow diagram was included to clearly illustrate the survey process (Fig 1).

Results

L152 to 156: This part belongs to the materials and methods section.

R. Moved

L185: The entire description in this paragraph belongs to table 2.

R. We beleive the review had the wrong Table for this comment. Could you please clarify and we will gladly perform the correction

Figures

Figure 1: It would be more didactic and easier for readers to understand if the authors presented the sequence of comparisons first HPxMP, followed by HPxLP, and MPxLP.

R. Corrected

Tables

Table 1: Herd ‘Her milk production level’

R. Corrected

Conclusion

The conclusion should be a little more straightforward. Remove the initial section summarizing the results; this has already been presented in the respective section.

R. The Conclusion was revised accordingly. The initial summary of results was removed, as these findings are presented in the Results section, and the Conclusion was restructured to enhance clarity and objectivity.

References

L486: Custom

R. Corrected

Reviewer #1: Introduction-

The Introduction is a jumble of sentences without a clear logic. What is the study about? What is the context that justifies the research? What is the research intended to answer? The Introduction should be rewritten to clarify the context, justification, state of the art, and objective of the study.

R. The Introduction was completely rewritten in response to the reviewer’s comment. The revised version now presents a clearer logical structure, including the study context, justification, current state of the art, and the specific objective of the research.

Line 53-54- "As a result, dairy production has grown substantially worldwide, where milk production per cow increased by 88% from 2017 to 2018." Where did milk production per cow increase? It's not possible that this happened worldwide.

R. There was a typographical error; however, due to the restructuring of the Introduction, the reference was removed.

Line 55-56- "In addition, from 2006 to 2017, fluid milk production increased by 45% country-wide and reached 35.4 billion L in 2020." Which country are you referring to?

R. It referred to Brazil; however, this reference was removed following the reformulation of the Introduction.

Material and Methods- The description of the research methodology is incomplete. For example, why was the survey only conducted in the southern and southeastern regions of Brazil?

R. Because these regions represent the majority of confined dairy systems in Brazil.. These regions are vast. Where were the producers who received the questionnaires concentrated (state and region)?

R. Survey responses were predominantly from Minas Gerais (n = 55; 67%), followed by Santa Catarina (n = 9; 11%), Rio Grande do Sul (n = 7; 9%), Paraná (n = 5; 6%), São Paulo (n = 4; 5%), and Espírito Santo and Rio de Janeiro (n = 1 each; 1%).

Were the state and region factors considered in the data analysis?

R. No. Only the production level was considered in the data analysis.

What was the process for developing the 38 questions in the questionnaire (based on the literature)?

R. Questions were developed based on the most common feeding practices and management strategies reported in the literature.

Was the questionnaire sent only once, or were there multiple attempts?

R. The questionnaire was sent to participants only once.

The content between lines 108 and 124 is the result of the questionnaires. Why are they included in the materials and methods section?

R. It was moved to results section

Line 39-140- This classification reflects standard benchmarks used in the region to describe small, medium, and large-scale dairy operations." What is the reference for this classification?

R. It was based on the expertise of researchers familiar with the Brazilian dairy sector, as there is no official classification of milk production levels for confined dairy systems in Brazil.

Results- The results are discussed superficially, descriptively, without critical analysis of the findings and their meaning.

Lines 153-155- 138 responses , 21 blank responses and 33 incomplete surveys were excluded from the dataset. The final analysis was conducted using the remaining 82 complete responses. The correct number is 84.

R. A typographical error occurred in reporting the number of incomplete surveys; the correct number is 35 rather than 33. Consequently, the total number of respondents was 82.

Lines 157-159. It is important to note that the number of respondents represents only a small fraction of the Brazilian dairy producer population, which totaled approximately 1.1 million in 2019." But this is the total number of producers in Brazil. It wasn't the study's target audience, but rather only those in confined systems.

R. It was reformulated to corrected describe.

Reviewer #2: The manuscript ‘Feed Management Practices Used for High-Producing Dairy Cows’ aligns well with the scope of PLOS ONE and provides valuable insights into management practices in Brazilian intensive dairy systems. However, minor revisions are required before the manuscript can be accepted. The following suggestions are offered to improve clarity and rigor:

Title: The title could be made more informative by including the country, at least, to provide greater precision. Dairy intensive systems in Brazil differ from those in other regions and have unique characteristics, as discussed in the manuscript.

R. The title was modified to address the reviewer’s comment.

Abstract:

L31: What was the criterion for selecting those 500? Please add a brief explanation to clarify.

R. That was the total number of producers using confined systems, as obtained from the dairy industry and nutritionists.

Introduction:

L54: Review this statement.

R. It was removed after introduction.

L56: Please provide an updated value for milk production in Brazil.

R. Updated as requested

L58-59: Please revise this sentence, as the phrasing currently resembles AI-generated text format. Ensure it aligns with a natural, academic writing style (review throughout the manuscript).

R. It was revised.

Lines 73-76: The stated objective seems too narrow. Based on the manuscript, the study does not focus solely on high milk production pens but includes herds across low, medium, and high production levels. Consider revising the objective to reflect the full scope of the study.

R. We appreciate the reviewer’s comment. Although farms in the study included herds with different production levels, the experimental unit of interest was intentionally restricted to high-producing pens. This was a deliberate design choice rather than a limitation. In Brazilian dairy systems, management practices are highly heterogeneous across pens within the same farm and are commonly stratified by milk yield. High-producing pens are typically prioritized for more intensive and higher-quality management strategies, including the specific practices evaluated in this study. Therefore, focusing on high-producing pens increased the likelihood that the targeted management interventions were consistently applied and biologically relevant. To address the reviewer’s concern, we have revised the objective to clarify that while farms represented a range of production levels, the study specifically evaluated management practices as implemented in high-producing pens, which are the pens most likely to receive and respond to the practices under investigation. This revision ensures that the stated objective accurately reflects the true scope and rationale of the study design.

Material and methods:

Start the section with approval from the Ethics Committee.

Done

L109: This information is duplicate. Please review. Revised

L110-124: Please review this text. In my opinion, it does not belong in the Materials and Methods section. Consider moving it to the Discussion or removing it.

R. It was moved to results section.

Results:

L164: Review last column format. Corrected.

L166: Standardize with others. Corrected.

L169: I did not understand. Corrected.

R. These values represented the number of respondents in each category. They were revised, deemed not relevant, and therefore removed.

L186-187: Blank lines. Corrected.

L191: A reference is not needed if this information comes from the present study.

R. The information does not come from the present study, and thus the reference was retained.

L224-229: Review the use of “;” and “.”. Corrected.

L236: No not use “numerically”. Removed.

Discussion:

It would be interesting to discuss the survey’s representativeness in terms of herd size and production level relative to the Brazilian dairy producer population.

L344: Blank line. Removed

L355-358: Avoid conclusion statements in each section. Removed

L376: Please rephrase. Done

L387: Review to improve readability. Revised

L408-410: Please review the text.

This result also deserves more discussion and implication “Across all HLMP groups, 83% of producers reported evaluating FE on a monthly basis”. Revised

L426-440: I suggest removing this section from the text, as the manuscript is already concise and includes a conclusion section. Revised and removed

Conclusion:

L444: No dot use methods or results stateme.nts in this section. Be objective. Revised to be more straightforward

Acknowledgements:

Here is a clean, journal-ready version:

New Financial Disclosure

This study did not receive specific funding for its execution. The Conselho Nacional de Desenvolvimento Científico e Tecnológico (CNPq, Brazil) provided an internship scholarship to the first author during their PhD program. This support was for academic training purposes only and did not fund the design, data collection, analysis, interpretation, or publication of this study.

Reviewer #3: The title should be "Feed management practices used for high-producing dairy cows in Brazil" Revised as suggested.

The manuscript provides important and up-to-date information. However, starting with the title, it should be clear that it refers to dairy systems in Brazil. Revised as suggested.

I recommend having all the manuscript writing and grammar reviewed by a native English speaker. As an example on the first page, there are more appropriate terms, for "furnish" such as "provide" on line 62, or "strategies employed on farms" such as "practices applied on farms" on line 63, and so on throughout the text.

R. The manuscript was thoroughly revised to ensure clarity, accuracy, and appropriate scientific English.

There is also incorrect information such as "As a result, dairy production has grown substantially worldwide, where milk production per cow increased by 88% from 2017 to 2018" line 53; Revised. It was removed from introduction after revision process based on suggestion from another reviewer

It should be clear in phrases like " In addition, from 2006 to 2017, fluid milk production increased by 45% country-wide and reached 35.4 billion L in 2020", especially in the introduction, that you are referring to Brazil. Revised. It was removed from introduction after revision process based on suggestion from another reviewer

---

## [Decision Letter · Decision Letter 1]

23 Feb 2026

Feed management practices used for dairy cows in confined dairies in Brazil

PONE-D-25-41297R1

Dear Dr. Marcelo B. Abreu,

We’re pleased to inform you that your manuscript has been judged scientifically suitable for publication and will be formally accepted for publication once it meets all outstanding technical requirements.

Kind regards,

Manasa Varra

Academic Editor

PLOS One

Additional Editor Comments (optional):

Dear Marcelo B. Abreu,

We are happy to inform that the revised manuscript, PONE-D-25-41297R1 entitled "Feed management practices used for dairy cows in confined dairies in Brazil" has addressed all the comments and hence is recommended for acceptance.

Reviewers' comments:

Reviewer's Responses to Questions

**Comments to the Author**

Reviewer #1: All comments have been addressed

Reviewer #2: All comments have been addressed

Reviewer #3: All comments have been addressed

2. Is the manuscript technically sound, and do the data support the conclusions?

Reviewer #1: Yes

Reviewer #2: Yes

Reviewer #3: Yes

3. Has the statistical analysis been performed appropriately and rigorously?

Reviewer #1: Yes

Reviewer #2: Yes

Reviewer #3: Yes

4. Have the authors made all data underlying the findings in their manuscript fully available?

Reviewer #1: Yes

Reviewer #2: Yes

Reviewer #3: No

5. Is the manuscript presented in an intelligible fashion and written in standard English?

Reviewer #1: Yes

Reviewer #2: Yes

Reviewer #3: Yes

Reviewer #1: (No Response)

Reviewer #2: The authors have done a good job revising the manuscript. The reviewers’ comments were carefully addressed, and the manuscript has improved in clarity, structure, and overall rigor. This study provides valuable benchmarking data on feed management practices in confined dairy systems in Brazil, relevant for producers, nutritionists, extension professionals, and policymaking within the dairy sector.

Reviewer #3: The authors cited that all data are fully available without restriction but state that the data will be made available in a public repository after the manuscript is accepted.

"Data will be make available in a public repository upon acceptance of the manuscript"

**Do you want your identity to be public for this peer review?** For information about this choice, including consent withdrawal, please see our Privacy Policy

Reviewer #1: No

Reviewer #2: No

Reviewer #3: No

---

## [Editor Report · Acceptance letter]

PONE-D-25-41297R1

PLOS One

Dear Dr. Marcondes,

I'm pleased to inform you that your manuscript has been deemed suitable for publication in PLOS One. Congratulations! Your manuscript is now being handed over to our production team.

Kind regards,

on behalf of

Dr. Manasa Varra

Academic Editor

PLOS One